# Circulating resistin and follistatin levels in obese and non-obese women with polycystic ovary syndrome: A systematic review and meta-analysis

**Tahereh Raeisi[1], Hossein Rezaie[2], Mina Darand[3], Akram Taheri[4], Nazila Garousi[5], Bahman Razi[6], Leonardo Roever[7], Reza Mohseni[8], Shimels Hussien Mohammed[9], Shahab Alizadeh[8]***

1 Department of Medicine, Hormozgan University of Medical Sciences, Bandar Abbas, Iran, 2 Department of Anatomy, School of Medicine Tehran University of Medical Sciences, Tehran, Iran, 3 Department of Clinical Nutrition and Dietetics, Faculty of Nutrition Sciences and Food Technology, National Nutrition and Food Technology Research Institute, Shahid Beheshti University of Medical Sciences, Tehran, Iran, 4 Department of Nutrition, Faculty of Medicine, Science and Research Branch, Islamic Azad University, Tehran, Iran, 5 Department of Clinical Nutrition, School of Nutrition and Food Science, Isfahan University of Medical Sciences, Isfahan, Iran, 6 Department of Hematology and Blood Banking, School of Allied Medical Sciences, Tehran University of Medical Sciences (TUMS), Tehran, Iran, 7 Department of Clinical Research, Federal University of Uberlândia, Minas Gerais, Brazil, 8 Department of Clinical Nutrition, School of Nutritional Sciences and Dietetics, Tehran University of Medical Sciences, Tehran, Iran, 9 Department of Community Nutrition, School of Nutritional Sciences and Dietetics, Tehran University of Medical Sciences-International Campus (TUMS-IC), Tehran, Iran

* sh_alizadeh@razi.tums.ac.ir

**Data Availability Statement:** All relevant data are within the manuscript and its Supporting Information files.

## Abstract

This meta-analysis was performed to resolve the inconsistencies regarding resistin and follistatin levels in women with polycystic ovary syndrome (PCOS) by pooling the available evidence. A systematic literature search using PubMed and Scopus was carried out through November 2020 to obtain all pertinent studies. Weighted mean differences (WMDs) with corresponding 95% confidence intervals (CIs) were calculated to evaluate the strength of the association between the levels of resistin and follistatin with PCOS in the overall and stratified analysis by obesity status. A total of 47 publications, 38 for resistin (2424 cases; 1906 controls) and 9 studies for follistatin (815 cases; 328 controls), were included in the meta-analysis. Resistin levels were significantly higher in PCOS women compared with non-PCOS controls (WMD = 1.96 ng/ml; 95%CI = 1.25–2.67, P≤0.001) as well as in obese PCOS women vs. obese controls, and in non-obese PCOS women compared with non-obese controls, but not in obese PCOS vs. non-obese PCOS patients,. A significantly increased circulating follistatin was found in PCOS patients compared with the controls (WMD = 0.44 ng/ml; 95%CI = 0.30–0.58, P≤0.001) and in non-obese PCOS women compared with non-obese controls and in obese PCOS women vs. obese controls, but, no significant difference in follistatin level was observed in obese PCOS compared with non-obese PCOS women. Significant heterogeneity and publication bias was evident for some analyses. Circulating levels of resistin and follistatin, independent of obesity status, are higher in women with PCOS compared with controls, showing that these adipokines may contribute to the pathology of PCOS.

**Funding:** The author(s) received no specific funding for this work.

**Competing interests:** The authors have declared that no competing interests exist.

## Introduction

Polycystic ovary syndrome (PCOS) is a common heterogeneous endocrine disease affecting about 10% of reproductive-aged women [1], which is featured by clustering of biochemical and clinical hyperandrogenemia, hirsutism, acne, oligo-or anovulation and polycystic ovaries [2]. Besides causing infertility, PCOS is closely related to obesity, insulin resistance (IR), type 2 diabetes, dyslipidemia, cardiovascular diseases, hepatic steatosis, and endometrial cancer [3–5], consequently leading to an adverse effect on health-related quality of life in PCOS women and a remarkable burden on the healthcare system [6]. Thus, the concept of PCOS involves multiple systems in the body and goes beyond the endocrinal and gynecological definitions to cover a broad array of conditions.

Up to date, although the exact pathogenesis and etiology of PCOS remain not fully understood, the phenotypic expression of patients with this syndrome are differentiated with some women being obese, some being normal weight, and some women presenting IR and elevated adipokines production [7]. However IR is not necessarily needed for the PCOS diagnosis, it occurs in approximately 50–80% of patients with this syndrome [8], and is considered to play a significant role in its etiology [2]. Regardless of body mass index (BMI), excess adiposity and elevated secretion of adipokines from adipocytes might be a linkage between the IR and ovulation disturbances [9]. The adipokine resistin, a 12.5 kDa cysteine-rich protein secreted by adipocytes, is identified as a strong candidate linking IR to excess adiposity [10]. Circulating resistin concentrations are significantly elevated in insulin-resistant mice and genetically or diet-induced obese mice [10, 11]. Moreover, resistin expression is reported to be up-regulated by dehydroepiandrosterone [12], proposing that resistin and androgen synthesis, a common condition in PCOS [13], may be related. In line with these findings, Seow et al. [14] found that upregulation of resistin might be involved in the pathogenesis of PCOS. Another adipokine that might play an important role in metabolic and endocrine complications in PCOS is follistatin, a member of the transforming growth factor-b superfamily [15], which acts as a significant regulator of follicular development and has been identified as a candidate gene for PCOS [16]. It is also recognized that an increase in follistatin inhibits follicle-stimulating hormone (FSH) production and might inhibit follicular development and increase the production of androgen by ovaries, both of which are fundamental in PCOS [17]. The authors still need to provide clear justification for selecting these resistin and follistatin particularly while there are many other important adipokines that have been studied in context of PCOS.

Recently, there has been an increasing interest in exploring the changes in circulating adipokines in PCOS patients; meta-analysis of changes in many important adipokines in context of PCOS such as omentin-1 [18], leptin [19, 20], interleukin-6(IL-6) [21], visfatin [22], tumor necrosis factor-alpha (TNF-a) [23], adiponectin [24], and retinol-binding protein 4 (RBP4) [25] has been performed previously. But, in spite of the potentially important metabolic roles of resistin and follistatin in PCOS and contradictory results of the previous studies, there is no meta-analysis on these hormones yet, indicating the importance of pooling the available data in this regard. Besides, for patients with PCOS stratified by weight status (obese versus lean), the results of individual studies regarding circulating resistin and follistatin were also inconsistent. Thus, the primary aim of the present systematic review and meta-analysis was to clarify the relation of circulating levels of resistin and follistatin to PCOS stratified by weight status. The secondary aim was to evaluate the changes in these adipokines in obese PCOS patients compared with non-obese PCOS patients.

## Materials and methods

This systematic review and meta-analysis was performed by following the Preferred Reporting Items for Systematic reviews and Meta-Analyses (PRISMA) statement [26].

## Search strategy

A comprehensive literature search was performed in PubMed and Scopus to obtain all pertinent human studies published up to November 2020. Following medical subject heading (MeSh) terms and text words were used: (resistin or adipose tissue-specific secretory factor or C/EBP-epsilon-regulated myeloid-specific secreted cysteine-rich protein or ADSF or XCP1 or follistatin or acti-vin-binding protein) and (PCOS or polycystic ovary syndrome). The search strategy was limited to articles published in English. Moreover, the reference lists of included studies and review articles were manually searched to obtain other possible relevant studies that may have been missed in the initial search. Two investigators (SHA and LR) independently searched the electronic databases and screened the titles/abstracts and full-text studies after excluding duplicated publications. Any disagreement in the screening process was resolved by involving a third reviewer.

## Inclusion and exclusion criteria

Studies were eligible for the current meta-analysis if they met the following criteria: a) studies should be published in English; b) reported circulating levels of resistin or follistatin in women with PCOS compared with healthy female controls; c) reported body mass index (BMI) or weight status of the participants; d) reported the total means of resistin or follistatin and standard deviations (SDs) or sufficient information to calculate them; e) included an acceptable diagnosis of PCOS based on the Rotterdam criteria, the Androgen Excess and PCOS Society criteria, or National Institute of Health (NIH) criteria. Only the most informative or the newest study was included when multiple studies were published based on the same population. Studies were excluded if they enrolled participants with diseases other than PCOS or were reviews, editorials, case reports, conference abstracts, cell line studies, animal studies, letters to editors, and studies without controls. EndNote program was used to facilitate the screening process. Two independent authors (SHA and LR) screened the titles/abstracts of studies to find whether studies are eligible for inclusion. If the abstracts of articles seemed pertinent, then the full-text assessment was done and the screening forms were filled out to select the eligible studies based on the inclusion criteria. Any disagreements were discussed and resolved by consensus with a third reviewer.

## Data extraction and quality assessment

Two investigators conducted data extraction and quality assessment of the included publications independently and dissimilarities were resolved by referring back to the original citations. The following information was extracted: first author's last name, country, year of publication, study design, age, numbers of cases and controls, ethnicity, BMI or weight status, and circulating resistin and follistatin levels (means and standard deviations). Because of the variety of definitions for obesity among the studies, it was defined as BMI above 25 or 30 kg/m 2. For articles providing the standard errors of means, the standard deviations (SDs) were calculated by multiplying the standard errors by the square roots of the sample size [27]. For publications reporting the medians and the corresponding interquartile range, medians were considered as means while the SDs were estimated by dividing the widths of the interquartile ranges by 1.35 [2]. The quality of the included studies was evaluated with the Newcastle-Ottawa Scale [28], which covers nine items, and scores range from 0 to 9. Studies with scores of ≤4, 5 to 6, and 7 or above are categorized as low-quality, medium-quality, and high-quality studies, respectively.

## Statistical analysis

The summary weighted mean difference (WMD) with 95%CIs was estimated for circulating levels of resistin and follistatin in PCOS cases versus controls (or between obese and non-

obese PCOS women). The $I^2$ statistic and Cochrane Q statistic were applied to assess the statistical heterogeneity across studies. A p value<0.1 for Q-statistic or $I^2$ larger than 50% was considered significant heterogeneity. When significant evidence for heterogeneity was detected, the random-effects model was used for analyses; otherwise, a fixed-effect model was applied. The sensitivity analysis was conducted by removing one study each time and recalculating WMD with a 95% confidence interval (CI) to find the effect of individual studies on the pooled effect sizes. Publication bias was evaluated using funnel plot asymmetry and Egger's linear regression method [29] and P values ≤0.05 were considered statistically significant. All analyses were done with the use of Stata software (Version 13.0; StataCorp, College Station, TX, USA).

## Results

### Search result and study characteristics

The initial search strategy found 870 studies, of which 805 studies were excluded after screening by titles/ abstracts or duplicate publications, and 65 studies were eligible for full-text assessment. Finally, a total of 47 articles, 38 for resistin (2424 cases; 1906 controls) [7, 9, 11, 14, 30–63] and 9 studies for follistatin (815 cases; 328 controls) [15, 17, 64–70], were included in the present meta-analysis based on the inclusion criteria. The flowchart summarizing the screening process is reported in Fig 1. The included studies were published between 2001 and 2020. The included studies also had subpopulations based on obesity status, which were analyzed separately. For resistin, there were 20 studies comparing circulating resistin in healthy non-obese PCOS women vs. non-obese control, 12 studies on obese PCOS vs. obese healthy women, and 18 studies on obese PCOS vs. non-obese PCOS patients [7, 9, 30, 32–34, 36–39, 44, 51, 53–55, 59–61]. Moreover, for follistatin, there were 4 publications comparing circulating follistatin in non-obese PCOS women vs. non-obese healthy controls [15, 65–67], 2 studies on obese PCOS vs. obese healthy women [17, 65], and 2 studies on obese PCOS vs. non-obese PCOS patients [65, 67]. The quality of studies was medium to high, with scores ranging from 4 to 8 (S1 Table).

The characteristics of the included articles are reported in Table 1.

### Quantitative synthesis of data

**Resistin levels.** In the pooled analysis of all eligible studies (38 studies), circulating resistin levels were significantly higher in PCOS women compared with non-PCOS controls (random effects, WMD = 1.96 ng/ml; 95% CI = 1.25 to 2.67, P ≤0.001) (Fig 2 and Table 2), with a significant heterogeneity across studies (I2 = 96.9%; p = ≤0.001). Resistin levels were also significantly higher in obese PCOS women vs. obese controls (random effects, WMD = 1.36 ng/ml; 95% CI = 0.53 to 2.20, P = 0.001) (Fig 3 and Table 2), and in non-obese PCOS women compared with non-obese controls (random effects, WMD = 1.60 ng/ml; 95% CI = 0.57 to 2.63, P = 0.002) (Fig 4 and Table 2), but not in obese PCOS vs. non-obese PCOS patients (Fig 5 and Table 2).

**Follistatin levels.** When all eligible studies were pooled (8 studies), a significantly increased levels of circulating follistatin were found in PCOS patients compared with the controls (random effects, WMD = 0.44 ng/ml; 95% CI = 0.30 to 0.58, P ≤0.001) (Fig 6 and Table 2); although, a significant evidence for heterogeneity was detected (I2 = 99.2%; p = ≤0.001). Follistatin levels were also significantly higher in non-obese PCOS women compared with non-obese controls (random effects, WMD = 0.64 ng/ml; 95% CI = 0.34 to 0.94, P ≤0.001) (S1 Fig and Table 2) and in obese PCOS women vs. obese controls (WMD = 0.58 ng/ml; 95% CI = 0.37 to 0.80, P ≤0.001) (S2 Fig and Table 2). No significant difference in

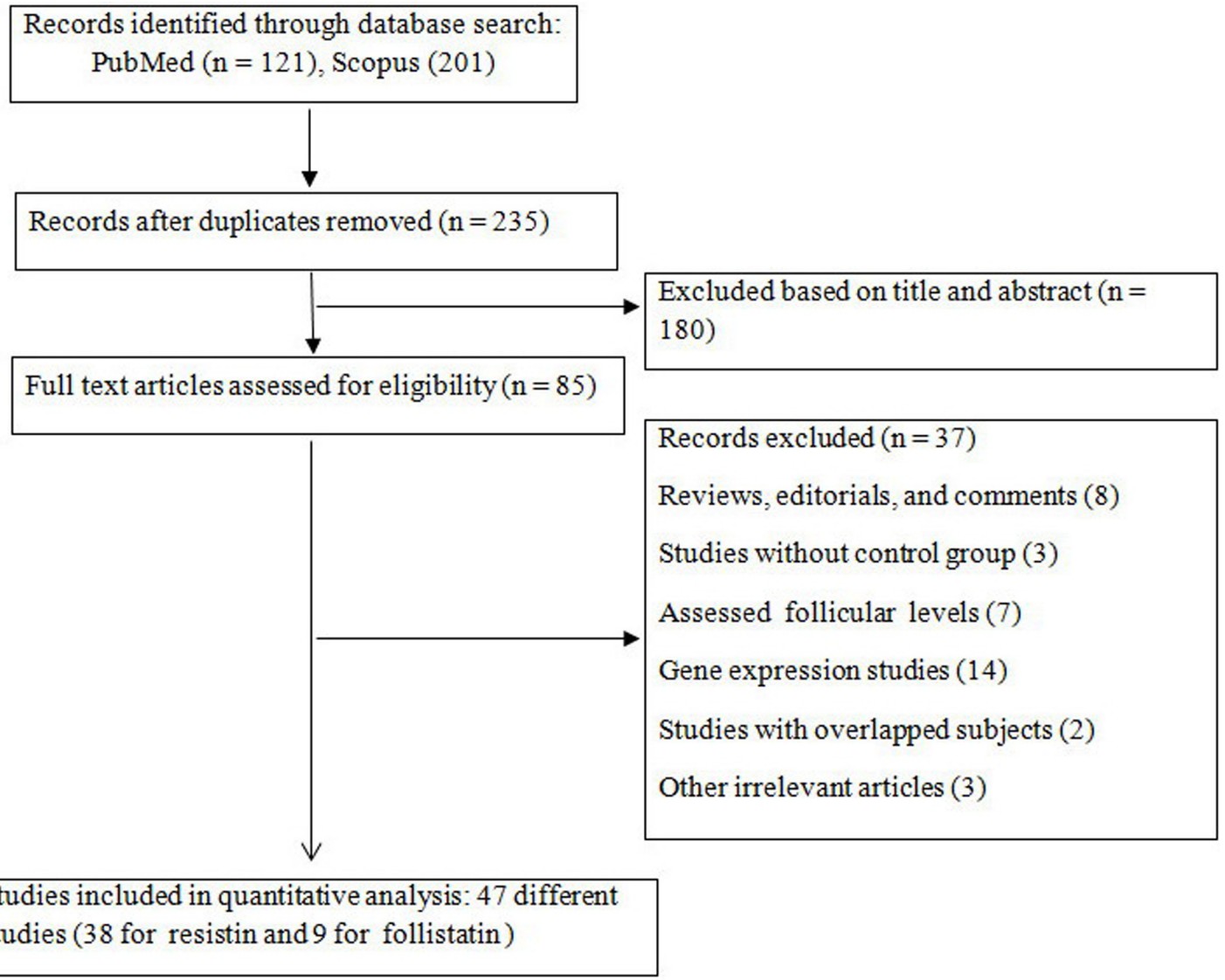

**Fig 1. Flow diagram of the literature search.**

follistatin level was observed in obese PCOS compared with non-obese PCOS women (S3 Fig and Table 2).

**Met-regression, sensitivity analysis, and publication bias.** Meta-regression analysis coefficients for serum resistin and follistatin levels in the examined group of studies showed that difference of mean age (Coefficient: 0.24, SE: 0.22, p = 0.30) and sample size (Coefficient: 0.003, SE: 0.003, p = 0.50) did not change the relation of resistin to PCOS. For follistatin, it was found that the relationship between follistatin and PCOS is modified by the difference of mean age between the PCOS patients and healthy controls (Coefficient: - 0.06, SE: 0.01, p = 0.007) so that by an increase in the difference of mean age between the PCOS patients and healthy controls, there was a significant reduction in the mean difference of follistatin between the groups (Fig 7). Sensitivity analysis was conducted by deleting the studies one by one in the meta-analysis each time to reflect the impact of the single studies on the pooled WMD. Finally, no single study considerably affected the pooled WMD for resistin (S2 Table) and follistatin (S3 Table),

**Table 1. The characteristics of the included studies in meta-analysis.**

| study | country | year | Outcome adipokine | Sample size | | Age (mean or range) | | Comparisons |
|---|---|---|---|---|---|---|---|---|
| | | | | PCOS | Control | PCOS | Control | |
| Panidis el al. | Greece | 2004 | resistin | 70 | 20 | 26.8 ± 5.2 | 28.6 ± 4.5 | PCOS vs. control |
| | | | | | | | | None-obese PCOS vs. None-obese control |
| | | | | | | | | Obese PCOS vs. none-obese PCOS |
| Seow el al. | Taiwan | 2004 | resistin | 17 | 10 | 28.9 ± 5.1 | 25.4 ± 4.3 | PCOS vs. control |
| LU el al. | China | 2005 | resistin | 20 | 20 | 29.68 ± 3.74 | 30.81 ± 2.74 | PCOS vs. control |
| | | | | | | | | None-obese PCOS vs. None-obese control |
| Seow el al. | Taiwan | 2005 | resistin | 21 | 18 | 23–40 | 24–44 | PCOS vs. control |
| | | | | | | | | None-obese PCOS vs. None-obese control |
| Morreale el al. | Spain | 2006 | resistin | 76 | 40 | 26 ± 6 | 31 ± 8 | PCOS vs. control |
| | | | | | | | | None-obese PCOS vs. None-obese control |
| | | | | | | | | Obese PCOS vs. obese control |
| | | | | | | | | Obese PCOS vs. none-obese PCOS |
| Bideci el al. | Turkey | 2008 | resistin | 28 | 19 | 15.15 ± 0.78 | 14.8 ± 1 | PCOS vs. control |
| | | | | | | | | Obese PCOS vs. obese control |
| | | | | | | | | Obese PCOS vs. none-obese PCOS |
| YILMAZ et al. | Turkey | 2009 | resistin | 76 | 46 | 24.1 ± 5.32 | 23.98 ± 6.08 | PCOS vs. control |
| | | | | | | | | None-obese PCOS vs. None-obese control |
| | | | | | | | | Obese PCOS vs. obese control |
| | | | | | | | | Obese PCOS vs. none-obese PCOS |
| ARIKAN et al. | Turkey | 2010 | resistin | 31 | 25 | 21.8 ± 5.4 | 24.9 ± 5.7 | PCOS vs. control |
| | | | | | | | | None-obese PCOS vs. None-obese control |
| Glinianowicz el al. | Poland | 2011 | resistin | 41 | 16 | 24.9 ± 6.5 | 27.8 ± 7.1 | PCOS vs. control |
| | | | | | | | | None-obese PCOS vs. None-obese control |
| | | | | | | | | Obese PCOS vs. none-obese PCOS |
| Glinianowicz el al. | Poland | 2013 | resistin | 87 | 67 | 25.4 ± 5.5 | 25.7 ± 4.9 | PCOS vs. control |
| | | | | | | | | None-obese PCOS vs. None-obese control |
| | | | | | | | | Obese PCOS vs. obese control |
| | | | | | | | | Obese PCOS vs. none-obese PCOS |
| Cassar el al. | Australia | 2015 | resistin | 44 | 40 | 28 ± 4.08 | 31.15 ± 6.5 | PCOS vs. control |
| | | | | | | | | None-obese PCOS vs. None-obese control |
| | | | | | | | | Obese PCOS vs. obese control |
| | | | | | | | | Obese PCOS vs. none-obese PCOS |
| Oz Gul el al. | Turkey | 2015 | resistin | 37 | 18 | 25.54 ± 4.8 | 29.8 ± 4.1 | PCOS vs. control |
| | | | | | | | | None-obese PCOS vs. None-obese control |
| | | | | | | | | Obese PCOS vs. none-obese PCOS |
| Nambiar el al. | India | 2016 | resistin | 282 | 200 | 28.64 ± 5.06 | 31.11 ± 5.13 | PCOS vs. control |
| | | | | | | | | None-obese PCOS vs. None-obese control |
| | | | | | | | | Obese PCOS vs. obese control |
| | | | | | | | | Obese PCOS vs. none-obese PCOS |
| CHEN el al. | Taiwan | 2007 | resistin | 15 | 29 | 32.33 ± 4.18 | 33.89 ± 4.27 | PCOS vs. control |
| | | | | | | | | None-obese PCOS vs. None-obese control |
| CHU el al. | China | 2009 | resistin | 35 | 40 | 28 ± 3 | 27 ± 4 | PCOS vs. control |
| WANG el al. | China | 2010 | resistin | 46 | 50 | 17–38 | NR | PCOS vs. control |
| | | | | | | | | None-obese PCOS vs. None-obese control |
| | | | | | | | | Obese PCOS vs. obese control |
| | | | | | | | | Obese PCOS vs. none-obese PCOS |

*(Continued)*

**Table 1.** (*Continued*)

| study | country | year | Outcome adipokine | Sample size | | Age (mean or range) | | Comparisons |
|---|---|---|---|---|---|---|---|---|
| | | | | PCOS | Control | PCOS | Control | |
| Sarray el al. | Bahrain | 2015 | resistin | 211 | 215 | 28.6 ± 6.1 | 27.5 ± 7 | PCOS vs. control |
| | | | | | | | | Obese PCOS vs. obese control |
| Yasar NAWAZ | Pakistan | 2020 | resistin | 40 | 7 | 24.20 ± 4.762 | 22.30 ± 3.517 | PCOS vs. control |
| Bertha Pangaribuan | Indonesia | 2011 | resistin | 24 | 18 | 20–40 | 22.2 ± 2.1 | PCOS vs. control |
| | | | | | | | | None-obese PCOS vs. None-obese control |
| | | | | | | | | Obese PCOS vs. none-obese PCOS |
| GUVEN | Turkey | 2010 | resistin | 22 | 16 | 15.2 ± 1 | 15.1 ± 1 | PCOS vs. control |
| | | | | | | | | None-obese PCOS vs. None-obese control |
| | | | | | | | | Obese PCOS vs. obese control |
| | | | | | | | | Obese PCOS vs. none-obese PCOS |
| M. Erkan | Turkey | 2014 | resistin | 28 | 28 | 22 ± 3.75 | 24 ± 4 | PCOS vs. control |
| Christian Obirikorang | Ghana | 2019 | resistin | 104 | 52 | 32.85 ± 4.25 | 31.63 ± 4.88 | PCOS vs. control |
| Nikolaos Spanos | Greece | 2012 | resistin | 60 | 48 | 25.4 ± 6.2 | 30.6 ± 6.3 | PCOS vs. control |
| Baldani | Croatia | 2019 | resistin | 151 | 95 | 26.5 ± 6 | 26.4 ± 2.7 | PCOS vs. control |
| | | | | | | | | None-obese PCOS vs. None-obese control |
| Behboudi-Gandevani | Iran | 2017 | resistin | 104 | 58 | 29.4 ± 5.3 | 31.8 ± 5.7 | PCOS vs. control |
| | | | | | | | | None-obese PCOS vs. None-obese control |
| | | | | | | | | Obese PCOS vs. obese control |
| | | | | | | | | Obese PCOS vs. none-obese PCOS |
| Mohd Ashraf Ganie | India | 2019 | resistin | 62 | 141 | 26.13 ± 4.43 | 26.57 ± 4.11 | PCOS vs. control |
| Farshchian | Iran | 2014 | resistin | 40 | 40 | 28.3 ± 5.1 | 28.3 ± 4.8 | PCOS vs. control |
| | | | | | | | | None-obese PCOS vs. None-obese control |
| | | | | | | | | Obese PCOS vs. obese control |
| | | | | | | | | Obese PCOS vs. none-obese PCOS |
| Atheer Mahde | Iraq | 2009 | resistin | 60 | 30 | 26.45 ± 4.65 | 28.87 ± 3.27 | PCOS vs. control |
| Nadine M. P. Daan | Netherland | 2016 | resistin | 68 | 64 | 28.5 ± 2.25 | 34.5 ± 1.75 | PCOS vs. control |
| Korczala | Poland | 2008 | resistin | 40 | 20 | 22 ± 2.5 | 21 ± 2.3 | PCOS vs. control |
| | | | | | | | | Obese PCOS vs. none-obese PCOS |
| Hung Shen | Taiwan | 2015 | resistin | 165 | 165 | 27 ± 5.7 | 28.6 ± 6.9 | PCOS vs. control |
| Baranova | USA | 2013 | resistin | 12 | 12 | 35.2 ± 9.60 | 37.6 ± 10.0 | PCOS vs. control |
| Çapoğlu | Turkey | 2009 | resistin | 45 | 20 | 23.8 ± 5 | 23.2 ± 3.25 | PCOS vs. control |
| Carmina | Italy | 2005 | resistin | 52 | 45 | 25.2 ± 1 | 25.1 ± 0.7 | PCOS vs. control |
| | | | | | | | | None-obese PCOS vs. None-obese control |
| | | | | | | | | Obese PCOS vs. obese control |
| | | | | | | | | Obese PCOS vs. none-obese PCOS |
| DİKMEN | TURKEY | 2010 | resistin | 55 | 49 | 23.72 ± 6.15 | 23.75 ± 4.5 | PCOS vs. control |
| | | | | | | | | None-obese PCOS vs. None-obese control |
| | | | | | | | | Obese PCOS vs. obese control |
| | | | | | | | | Obese PCOS vs. none-obese PCOS |
| WANG | China | 2012 | resistin | 70 | 35 | 26.1 ± 5.2 | 26.8 ± 4.7 | PCOS vs. control |
| | | | | | | | | None-obese PCOS vs. None-obese control |
| | | | | | | | | Obese PCOS vs. none-obese PCOS |
| Yılmaz | Turkey | 2005 | resistin | 40 | 20 | 46.38 ± 7.95 | 48.12 ± 9.02 | PCOS vs. control |
| Munir | USA | 2005 | resistin | 45 | 74 | 30 ± 5.366 | 31.6 ±7.74 | PCOS vs. control |

(*Continued*)

**Table 1.** (Continued)

| study | country | year | Outcome adipokine | Sample size | | Age (mean or range) | | Comparisons |
|-------|---------|------|-------------------|-------|---------|-------|---------|-------------|
| | | | | PCOS | Control | PCOS | Control | |
| Geva el al. | Israel | 2001 | follistatin | 44 | 62 | 29.4 ± 1 | 30.5 ± 1.2 | PCOS vs. control |
| | | | | | | | | None-obese PCOS vs. None-obese control |
| | | | | | | | | Obese PCOS vs. none-obese PCOS |
| Chen el al. | Taiwan | 2012 | follistatin | 239 | 38 | 26.9 ±0.5 | 26.3 ± 2.1 | PCOS vs. control |
| | | | | | | | | None-obese PCOS vs. None-obese control |
| | | | | | | | | Obese PCOS vs. none-obese PCOS |
| Teede el al. | Australia | 2013 | follistatin | 51 | 25 | 32.6 ± 0.8 | 36.4 ± 1.7 | PCOS vs. control |
| | | | | | | | | Obese PCOS vs. obese control |
| Norman el al. | Australia | 2001 | follistatin | 108 | 20 | 33.9 ± 1.2 | 29.4 ± 0.5 | PCOS vs. control |
| Shen el al. | China | 2004 | follistatin | 35 | 26 | 26.9 ± 3.8 | 28.1 ± 3.5 | PCOS vs. control |
| | | | | | | | | None-obese PCOS vs. None-obese control |
| Chen el al. | Taiwan | 2009 | follistatin | 155 | 37 | 24.5 ± 0.41 | 33.5 ± 0.45 | PCOS vs. control |
| | | | | | | | | None-obese PCOS vs. None-obese control |
| Köninger | Germany | 2018 | follistatin | 32 | 25 | 30 ± 4.93 | 31.64 ± 4.7 | PCOS vs. control |
| Adnan Kensara | Saudi Arabia | 2018 | follistatin | 63 | 65 | 31.6 ± 6.4 | 30.4 ± 5.2 | PCOS vs. control |
| Suganthi | India | 2010 | follistatin | 88 | 30 | NR | NR | PCOS vs. control |

NR: Not reported

showing that the results of this meta-analysis were statistically robust. A significant evidence for publication bias was detected by Egger's test for studies on resistin (t = 2.14, p = 0.03) (Fig 8 and Table 2).

## Discussion

Recently, changes in circulating levels of resistin and follistatin in women with PCOS have been investigated; although, the results of published studies are still inconclusive. Thus, this meta-analysis was performed to comprehensively analyze circulating levels of these adipokines in PCOS women compared with healthy women by considering weight status as an interacting factor. The results revealed that, overall, resistin and follistatin levels were significantly higher in PCOS women compared with healthy controls. The increased resistin and follistatin levels in PCOS patients were independent of obesity status.

In agreement with our findings, Pandis et al. [30] found that the resistin circulating levels were higher in patients with PCOS. They also found that subjects with a BMI≥25 kg/m2 compared with non-obese PCOS and non-obese healthy women have higher resistin levels. However, no difference in resistin levels was reported between patients with PCOS and a BMI<25 kg/m2 and healthy lean women [30], proposing that resistin may have a role in PCOS. Resistin has been reported as a potential link between insulin resistance and obesity [71]. The mRNA level of resistin has been revealed to be down-regulated by antidiabetic thiazolidinediones medicines, which bind to peroxisome proliferator-activated receptor-g (PPAR-g) in fat cells [10]. Resistin expression in adipocytes is increased in morbidly obese individuals compared with lean subjects [72]. In addition, in the adipocytes from women with PCOS, the mRNA level of resistin is 2-fold higher than that in healthy women [14]. It has been identified that there is a positive correlation between serum resistin and free testosterone levels [62]. Increased resistin concentration might increase ovarian androgen production in patients with

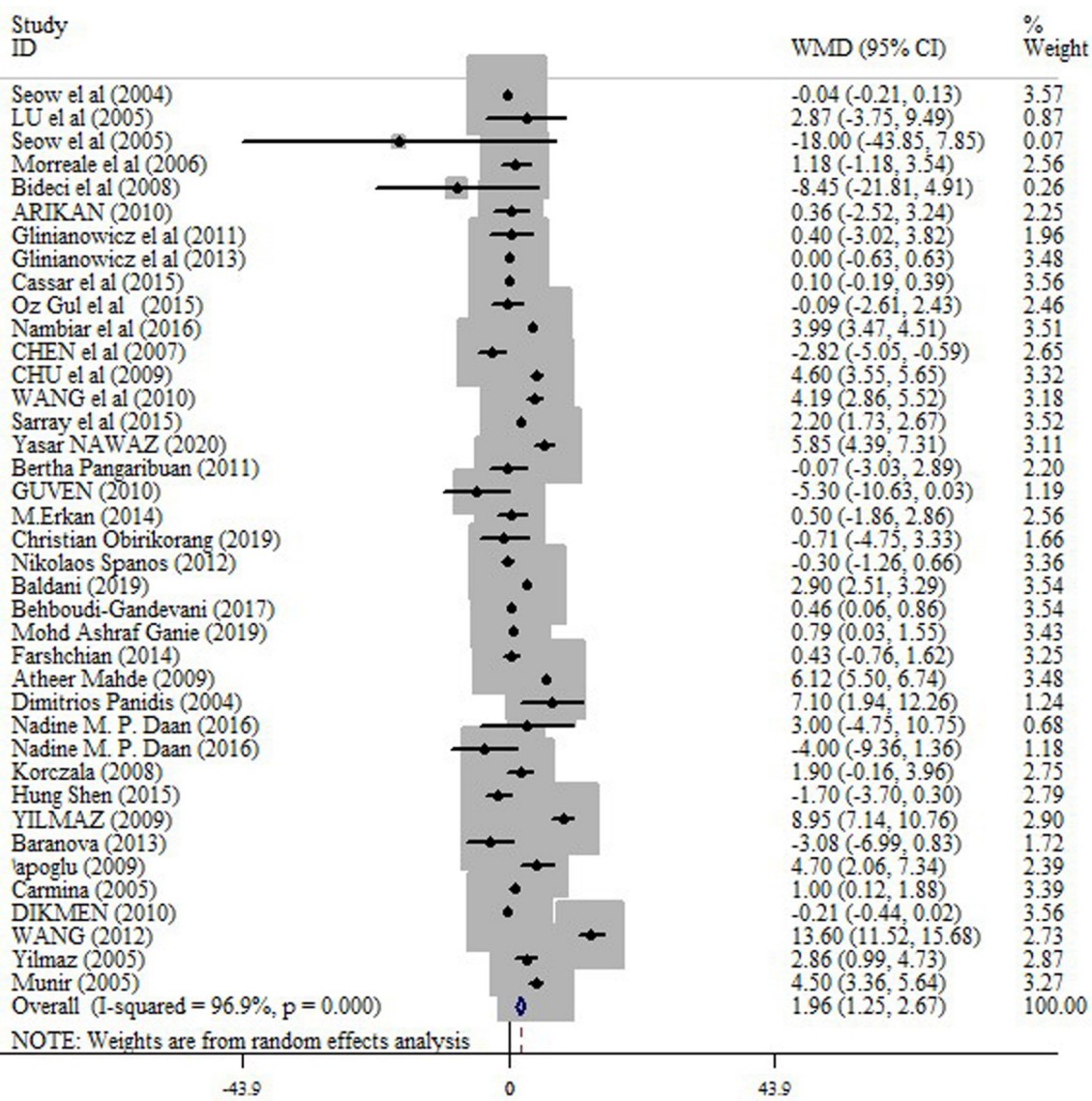

**Fig 2. Forest plot for the circulating resistin in PCOS women compared with healthy controls.**

PCOS [34]. Taken together, these data and our findings, show that resistin may contribute to PCOS and its clinical manifestations.

In line with our meta-analysis, previous studies have reported that circulating levels of follistatin are increased in PCOS independent of body mass index [64, 65], suggesting that obesity is not the explanatory factor for the increased levels of follistatin among women with PCOS. The source of circulating follistatin in women is unknown; nevertheless, this protein is released from ovarian granulosa cells in response to FSH stimulation [73]. An increase in the expression of follistatin is a possible mechanism in which follicular growth is inhibited [64]. These findings suggest that changes in the secretion of follistatin from the ovary or other organs might be involved in the pathophysiology of PCOS and describe the alteration in circulation levels of this protein. This is in line with this observation that PCOS is most closely related to areas adjacent to the follistatin gene in PCOS siblings [16]; however, later studies have

**Table 2. Meta-analysis of circulating resistin and follistatin in patients with PCOS.**

| Adipokine | Comparisons | Studies | Test of difference | | Heterogeneity | | Publication bias (Egger's) | |
|---|---|---|---|---|---|---|---|---|
| | | | WMD (95%CI) | p | $I^2$ (%) | P | t | P |
| Resistin | PCOS vs. control | 38 | 1.96 (1.25 to 2.67) | ≤0.001 | 96.9 | ≤0.001 | 2.14 | 0.03 |
| | None-obese PCOS vs. None-obese control | 20 | 1.60 (0.57 to 2.63) | 0.002 | 95.3 | ≤0.001 | 0.68 | 0.50 |
| | Obese PCOS vs. obese control | 12 | 1.36 (0.53 to 2.20) | 0.001 | 89.2 | ≤0.001 | 1.98 | 0.07 |
| | Obese PCOS vs. none-obese PCOS | 18 | 0.37 (-0.62 to 1.36) | 0.46 | 92.1 | ≤0.001 | 0.23 | 0.81 |
| Follistatin | PCOS vs. control | 9 | 0.44 (0.30 to 0.58) | ≤0.001 | 99.2 | ≤0.001 | 2.21 | 0.06 |
| | None-obese PCOS vs. None-obese control | 4 | 0.64 (0.34 to 0.94) | ≤0.001 | 78.1 | 0.003 | -0.17 | 0.88 |
| | Obese PCOS vs. obese control | 2 | 0.58 (0.37 to 0.80) | ≤0.001 | 0.0 | 0.40 | - | - |
| | Obese PCOS vs. none-obese PCOS | 2 | 0.12 (-0.11 to 0.34) | 0.29 | 0.0 | 0.84 | - | - |

identified the relationship to be much weaker than what previously thought [74]. Interestingly, in some studies, no significant difference in the level of follistatin was observed in the follicular fluid from the polycystic and healthy ovaries [75, 76]. The previous reports that failed to find any difference in the level of follistatin in the follicular fluids could be explained by thecal secretin of this adipokine and the rich blood supply in this region of the ovary resulting in rapid clearance of follistatin into the blood [64]. Moreover, these studies indicate the possibility of the extragonadal origin of the high circulating follistatin levels in women with PCOS. Since both resistin and follistatin are involved in clinical manifestations of PCOS, our findings are clinically important as they could be used as possible therapeutic targets in women with PCOS.

This meta-analysis resolved the inconsistency among the results of the previous studies; these findings are clinically important as an increase in resistin and follistatin observed in the present meta-analysis has been related to adverse metabolic consequences in individuals with PCOS. Higher levels of follistatin and resistin are associated with the lack of pre-ovular follicle development in PCOS or insulin resistance [64, 77]. Thus, the modulation of these hormones may be important therapeutic targets for new drugs in patients with PCOS. Based on the results of this study, follistatin and resistin could be studied in large cohort of PCOS patients

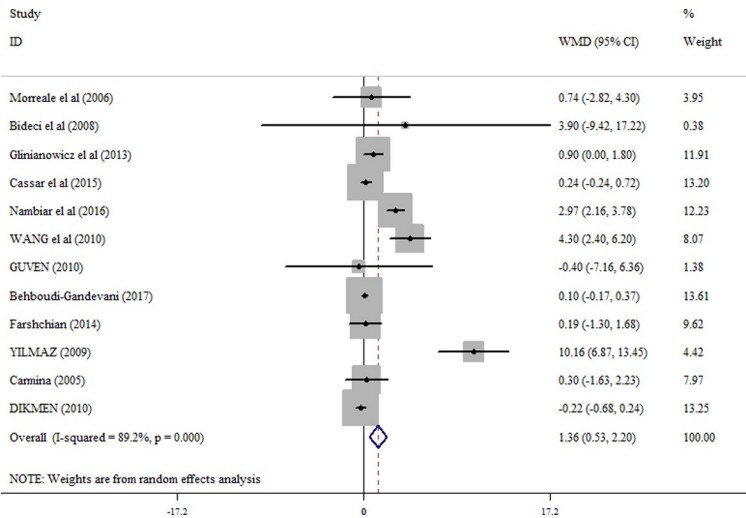

**Fig 3. Forest plot for the circulating resistin in obese PCOS women vs. obese controls.**

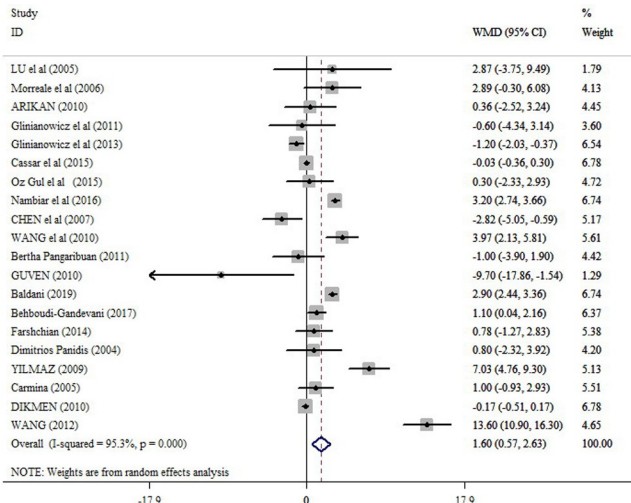

**Fig 4. Forest plot for the circulating resistin in non-obese PCOS women compared with non-obese controls.**

to better understand the pathophysiology of this disease. Also, future clinical trials implement interventions to improve circulating/follicular concentrations of these hormones.

To the best of our knowledge, this is the first meta-analysis summarizing available evidence to assess whether there is a relationship between circulating resistin and follistatin levels with PCOS. However, some limitations of the present meta-analysis should be acknowledged. First, the leading limitation of the current study is the remarkable heterogeneity across the included studies, which might decrease the reliability of our results. This heterogeneity may reflect clinical heterogeneity related to the differences in the accuracy and sensitivity of various methods applied to assess circulating levels of investigated adipokines, and differences in geographical regions, physical activity, diet, ethnicity of participants, and concomitant subclinical inflammatory diseases. Furthermore, some included studies had small sample sizes and the background of patients was different, which would lead to low statistical power and inconclusive conclusions among the studies. In the meta-regression analysis, we found that difference of

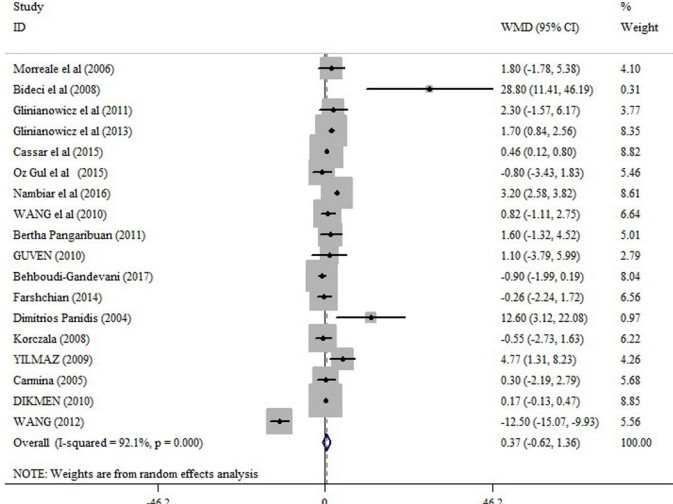

**Fig 5. Forest plot for the circulating resistin in obese PCOS vs. non-obese PCOS patients.**

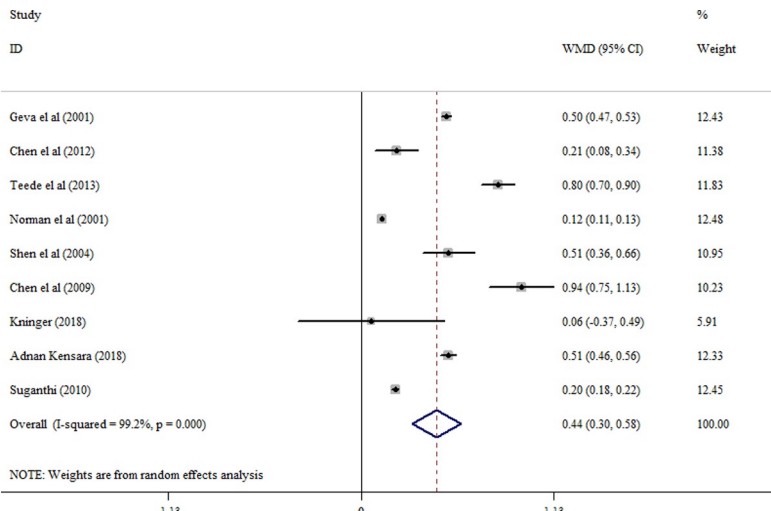

**Fig 6. Forest plot for the circulating follistatin in PCOS women compared with healthy controls.**

mean age is a source of observed heterogeneity among the studies. Nevertheless, in each individual study, cases and controls were matched for age. Our meta-regression was an exploratory analysis to assess whether difference of mean age between the PCOS patients and healthy controls across different studies affects pooled effect sizes despite matching cases and controls for age within studies. We revealed that the relationship between follistatin and PCOS is modified by mean difference of age between the PCOS patients and healthy controls, suggesting that results of studies with larger difference of mean age between the PCOS patients and healthy

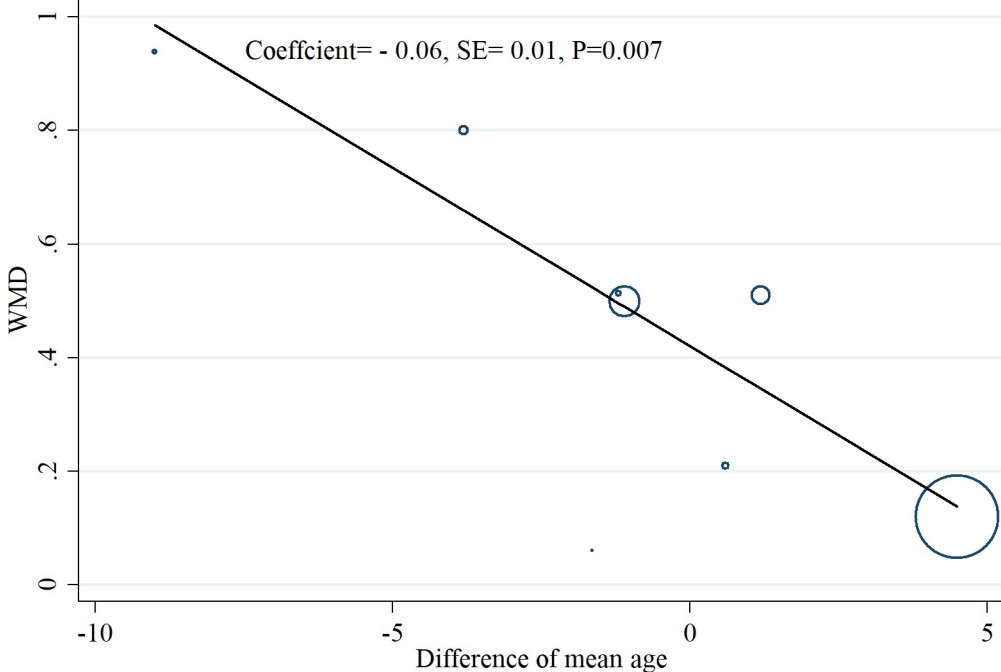

**Fig 7. Meta-regression analysis coefficients for serum follistatin levels in the examined group of studies based on the difference of mean age between the PCOS patients and healthy controls.**

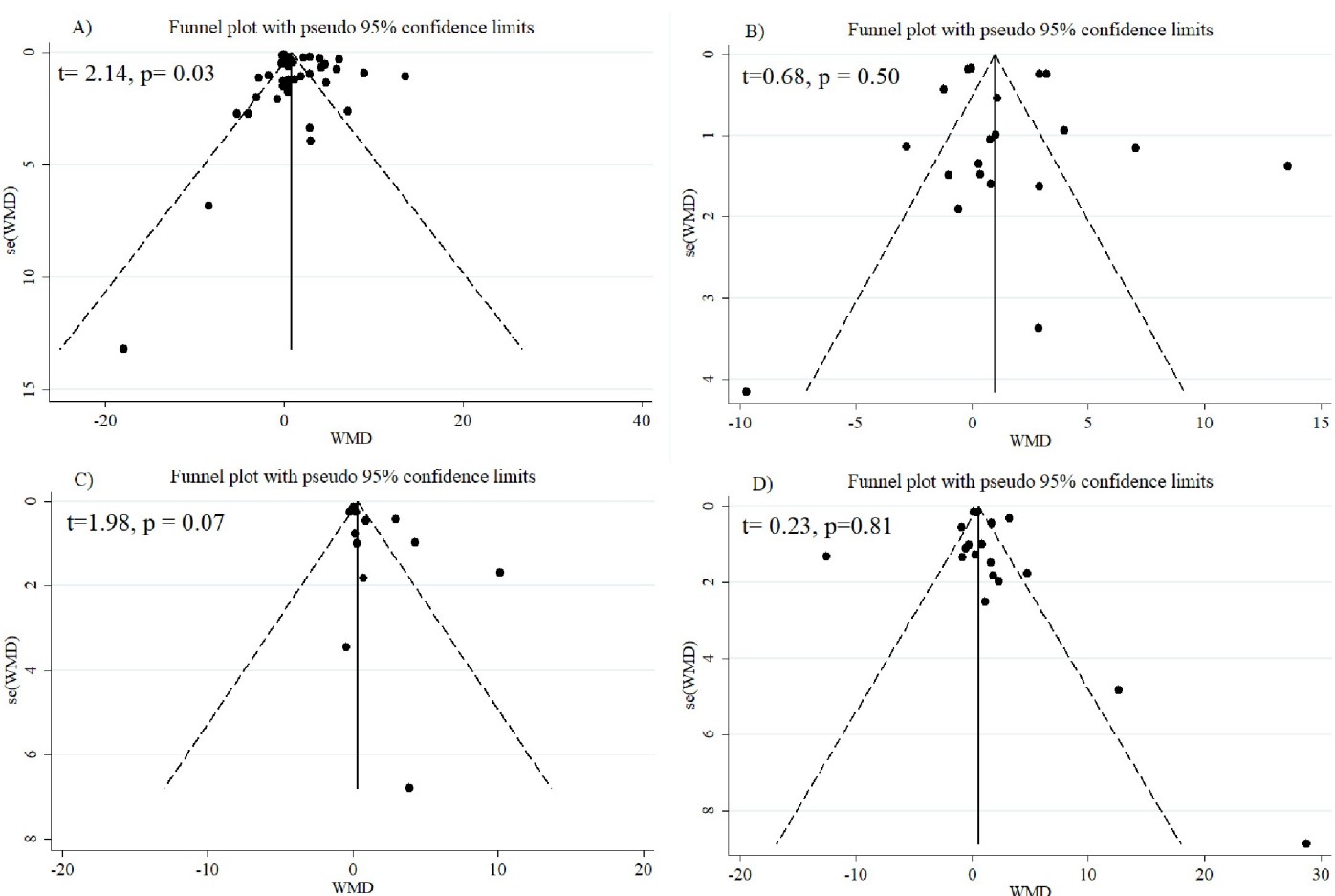

**Fig 8.** Funnel plots for publication bias for resistin in PCOS vs. control (A), none-obese PCOS vs. none-obese control (B), obese PCOS vs. obese control (C), and obese PCOS vs. none-obese PCOS (D).

controls should be interpreted cautiously. Second, the language restriction to English publications may increase the odds of publication bias. Third, the results were based on crude estimates without any adjustment for potential confounders, while a more accurate assessment should consider the confounding factors such as alcohol use, smoking status, dietary patterns, and environmental factors. It is noteworthy that cases and controls were matched for most important confounders such as age and BMI, but, similar to other epidemiological studies, residual confounding could not be ignored.

In conclusion, this meta-analysis suggested that resistin and follistatin levels, independent of obesity status, were higher in women with PCOS compared with those in healthy controls. Hence, resistin and follistatin might play important roles in the development of PCOS and may be useful biomarkers for the treatment of this disorder. Further well-designed studies with large sample sizes should be performed to examine the circulating levels of resistin and follistatin and their role in PCOS.

## Supporting information

**S1 Checklist. PRISMA 2009 checklist.**
(DOC)

**S1 Fig. Forest plot for the circulating follistatin in non-obese PCOS women compared with non-obese controls.**
(DOCX)

**S2 Fig. Forest plot for the circulating follistatin in obese PCOS women vs. obese controls.**
(DOCX)

**S3 Fig. Forest plot for the circulating follistatin in obese PCOS compared with non-obese PCOS women.**
(DOCX)

**S1 Table. Quality assessment of studies included in this systematic review and meta-analysis according to the Newcastle-Ottawa Scale (NOS) criteria.**
(DOCX)

**S2 Table. Sensitivity analysis by omitting single studies for studies investigating resistin levels in PCOS compared with healthy control women.**
(DOCX)

**S3 Table. Sensitivity analysis by omitting single studies for studies investigating follistatin levels in PCOS compared with healthy control women.**
(DOCX)

## Author Contributions

**Conceptualization:** Tahereh Raeisi, Nazila Garousi.

**Data curation:** Tahereh Raeisi, Mina Darand, Nazila Garousi.

**Formal analysis:** Hossein Rezaie.

**Investigation:** Bahman Razi.

**Methodology:** Hossein Rezaie, Akram Taheri, Leonardo Roever.

**Software:** Hossein Rezaie, Mina Darand, Leonardo Roever.

**Supervision:** Shahab Alizadeh.

**Validation:** Mina Darand, Reza Mohseni.

**Writing – original draft:** Tahereh Raeisi, Reza Mohseni, Shahab Alizadeh.

**Writing – review & editing:** Shimels Hussien Mohammed.

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
