## [Decision Letter · Decision Letter 0]

21 Aug 2020

PONE-D-20-19835

Circulating resistin and follistatin levels in obese and non-obese women with polycystic ovary syndrome: a systematic review and meta-analysis

PLOS ONE

Dear Dr. alizadeh,

Thank you for submitting your manuscript to PLOS ONE. After careful consideration, we feel that it has merit but does not fully meet PLOS ONE’s publication criteria as it currently stands. Therefore, we invite you to submit a revised version of the manuscript that addresses the points raised during the review process.

The reviewers have raised several concerns regarding the statistical analysis and the interpretation of the data to the published literature. In addition, one of the reviewers has highlighted that studies have been omitted from the analysis and these need to be identified and either included or a supplementary table detailing why they were excluded

We look forward to receiving your revised manuscript.

Kind regards,

Stephen L Atkin, MD

Academic Editor

PLOS ONE

Journal Requirements:

Reviewers' comments:

Reviewer's Responses to Questions

**Comments to the Author**

1. Is the manuscript technically sound, and do the data support the conclusions?

Reviewer #1: Yes

Reviewer #2: Partly

2. Has the statistical analysis been performed appropriately and rigorously? 

Reviewer #1: Yes

Reviewer #2: Yes

3. Have the authors made all data underlying the findings in their manuscript fully available?

Reviewer #1: Yes

Reviewer #2: Yes

4. Is the manuscript presented in an intelligible fashion and written in standard English?

Reviewer #1: No

Reviewer #2: Yes

5. Review Comments to the Author

Reviewer #1: The authors have done very good comprehensive analysis of the literature to clarify the role of resistin and follistatin in insulin resistant PCOS women. The methodology and statistical methods used are correct, however there are some concerns with regards to the style of presentation in the manuscript as detailed below.

Minor corrections:

1. In abstract line 12 full stop not required after ((PCOS).by )

2. Typo errors are present throughout the manuscript this needs to be corrected.

3. Multiple words are joined throughout the manuscript. Please get the manuscript proofread and correct them.

4. Un-necessary use of capital letter, too many parentheses/round brackets for example page 4 line 80-82 remove them.

5. Referencing within manuscript some places authors have used round brackets and other times they use box brackets use one type and be consistent.

6. Although authors mention that reason for selecting resistin and follistatin for this meta-analysis is to clarify role of these proteins in PCOS. Since contradicting/inconclusive reports are published in literature. The authors still need to provide clear justification for selecting these resistin and follistatin particularly while there are many other important adipokines that have been studied in context of PCOS.

7. Page 3 line 50 when describing resistin authors start by saying “The adipokine resistin, new 12.5 kDa cystine-rich protein….. The word new should be removed resistin was first discovered in 2001 almost 20 years back.

8. The authors have only selected 39 papers for this meta-analysis hence smaller population size. While there are many more relevant papers published on resistin, follistatin and PCOS. The authors should consider increasing the number of studies to get robust analysis.

9. Some of the sentences used in the manuscript are too long and difficult to follow for example page 10 first and second paragraph it is too long. The authors need to break them into smaller sentences so readers can follow them easily.

10. Move figure and table legends to end of the manuscript just before references.

11. This meta-analysis does not consider confounders which may influence their findings. However, the author acknowledges this as the limitation of the study.

12. Already people have reported that follistatin and resistin increases in PCOS subjects. Authors should write a small paragraph on benefits of this meta-analysis of this meta-analysis. Does this study allow the researchers to study follistatin and resistin in large cohort of PCOS subjects etc…??

Reviewer #2: This study summarizes the difference of circulating resistin and follistatin levels in obese and non-obese women with polycystic ovary syndrome using meta-analysis. I have below comments and questions.

Please show funnel plots for those meta-analysis of 10 or more studies.

There are several studies with big negative WMD, please report detail results for WMD from sensitivity analysis with forest plots or in tables.

What is the “mean difference of age”? Is it actually the difference between the mean age of the PCOS patients and the mean age of the healthy controls? If yes, you may use the term of “difference of mean age”.

In meta-regression, what is the purpose to test the effect of difference of mean age between the PCOS patients and healthy controls? In each study, does it test if ages are comparable between

PCOS patients and healthy controls? If they are not comparable, was the age adjusted for the reported difference of resistin and follistatin levels between PCOS and controls? These issues should be raised and discussed rather than simply testing if WMD is associated with the mean difference of age between the PCOS patients and healthy controls.

Line s 189-191, “For follistatin, it was found that the relationship between follistatin and PCOS is modified by mean difference of age between the PCOS patients and healthy controls (Coefficient: - 0.06, SE: 0.01, p=0.007)”. How will you apply this result? Would you conclude that results are questionable/unreliable from those studies with larger difference of mean age between the PCOS patients and healthy controls?

Lines 191-193, “…by increase in the mean difference of age between the PCOS patients and healthy controls, there was a significant elevation in the mean difference of follistatin between the groups (Fig 4).” This sentence indicates a positive correlation. But from Fig. 4, we see a negative correlation,

6. PLOS authors have the option to publish the peer review history of their article (what does this mean?). If published, this will include your full peer review and any attached files.

Reviewer #1: No

Reviewer #2: No

---

## [Author Response · Author response to Decision Letter 0]

30 Dec 2020

27 December 2020

Re:

MS title: " Circulating resistin and follistatin levels in obese and non-obese women with polycystic ovary syndrome: a systematic review and meta-analysis (Ms# 

PONE-D-20-19835)" 

Dear Editor-in-Chief of PLOS ONE

We appreciate the time and efforts by the editor and referees in reviewing our manuscript. Below are the reviewers’ verbatim comments. All changes in the manuscript have been highlighted with red font. 

We hope that with these changes and clarifications, our manuscript will be acceptable for publication. Nevertheless, we are prepared to revise our manuscript further, should it be necessary. 

COMMENTS FOR THE AUTHOR:

Reviewer #1

The authors have done very good comprehensive analysis of the literature to clarify the role of resistin and follistatin in insulin resistant PCOS women. The methodology and statistical methods used are correct, however there are some concerns with regards to the style of presentation in the manuscript as detailed below.

Minor corrections:

1. In abstract line 12 full stop not required after ((PCOS).by )

Response: done (see line 13)

2. Typo errors are present throughout the manuscript this needs to be corrected.

Response: Typo errors were revised throughout the manuscript.

3. Multiple words are joined throughout the manuscript. Please get the manuscript proofread and correct them.

Response: done

4. Un-necessary use of capital letter, too many parentheses/round brackets for example page 4 line 80-82 remove them.

Response: revised (see lines 88-93)

5. Referencing within manuscript some places authors have used round brackets and other times they use box brackets use one type and be consistent.

Response: References were presented consistently in the revised version of the manuscript. 

6. Although authors mention that reason for selecting resistin and follistatin for this meta-analysis is to clarify role of these proteins in PCOS. Since contradicting/inconclusive reports are published in literature. The authors still need to provide clear justification for selecting these resistin and follistatin particularly while there are many other important adipokines that have been studied in context of PCOS.

Response: done (see lines 67-76).

7. Page 3 line 50 when describing resistin authors start by saying “The adipokine resistin, new 12.5 kDa cystine-rich protein….. The word new should be removed resistin was first discovered in 2001 almost 20 years back.

Response: done (see line 51).

8. The authors have only selected 39 papers for this meta-analysis hence smaller population size. While there are many more relevant papers published on resistin, follistatin and PCOS. The authors should consider increasing the number of studies to get robust analysis.

Response: Many thanks for your knowledgeable comment. We updated the search and found a total 7 additional studies on resistin (references 57 to 63) and 1 study on follistatin (reference 70). Notably, there are many other studies on resistin, follistatin in PCOS patients that were not eligible for our meta-analysis as they assessed follicular levels or gene expression of these hormones or had not a control group. A total of 47 publications, 38 for resistin and 9 studies for follistatin, were included in the revised version of the manuscript, enabling us to reach a more robust conclusion (see Table 1, Table 2, and supplementary Table 1)

9. Some of the sentences used in the manuscript are too long and difficult to follow for example page 10 first and second paragraph it is too long. The authors need to break them into smaller sentences so readers can follow them easily.

Response: done (see lines 220-236).

10. Move figure and table legends to end of the manuscript just before references.

Response: Based on the journal guidelines, it is suggested to include figure and table legends within the manuscript. Nevertheless, in the revised version they were moved to end of the manuscript just before references.

11. This meta-analysis does not consider confounders which may influence their findings. However, the author acknowledges this as the limitation of the study.

Response: It is noteworthy that cases and controls were matched for most important confounders such as age and BMI, but, similar to other epidemiological studies, residual confounding could not be ignored. This is unavoidable in all meta-analyses. This was added to the manuscript (see lines 290-292). 

12. Already people have reported that follistatin and resistin increases in PCOS subjects. Authors should write a small paragraph on benefits of this meta-analysis of this meta-analysis. Does this study allow the researchers to study follistatin and resistin in large cohort of PCOS subjects etc…??

Response: the results of the previous studies were inconclusive. This meta-analysis for the first time resolved the inconsistency among the results of the previous studies; these findings are clinically important as an increase in resistin and follistatin observed in the present meta-analysis has been related to adverse metabolic consequences in individuals with PCOS. Higher levels of follistatin and resistin are associated with the lack of pre-ovular follicle development in PCOS or insulin resistance. Thus, the modulation of these hormones may be important therapeutic targets for new drugs in patients with PCOS. Based on the results of this study, follistatin and resistin could be studied in large cohort of PCOS patients to better understand the pathophysiology of this disease. Also, future clinical trials implement interventions to improve circulating/follicular concentrations of these hormones (see lines 264-272). 

Reviewer #2: This study summarizes the difference of circulating resistin and follistatin levels in obese and non-obese women with polycystic ovary syndrome using meta-analysis. I have below comments and questions.

1-Please show funnel plots for those meta-analysis of 10 or more studies.

Response: done (see fig 3, fig 4, and fig 5).

2-There are several studies with big negative WMD, please report detail results for WMD from sensitivity analysis with forest plots or in tables.

Response: Sensitivity analysis was conducted by deleting the studies one by one in the meta-analysis each time to reflect the impact of the single studies on the pooled WMD. No single study considerably affected the pooled WMD for resistin (Supplemental Table 2) and follistatin (Supplemental Table 3), showing that the results of this meta-analysis were statistically robust (see lines 203-207).

3-What is the “mean difference of age”? Is it actually the difference between the mean age of the PCOS patients and the mean age of the healthy controls? If yes, you may use the term of “difference of mean age”.

Response: “mean difference of age” represents the difference between the mean age of the PCOS patients and the mean age of the healthy controls. It was replaced with “difference of mean age” (see lines 196, 199, 201, and 284).

4-In meta-regression, what is the purpose to test the effect of difference of mean age between the PCOS patients and healthy controls? In each study, does it test if ages are comparable between

PCOS patients and healthy controls? If they are not comparable, was the age adjusted for the reported difference of resistin and follistatin levels between PCOS and controls? These issues should be raised and discussed rather than simply testing if WMD is associated with the mean difference of age between the PCOS patients and healthy controls.

Response: In each individual study, cases and controls were matched for age, thus ages were comparable between PCOS patients and healthy controls. Our meta-regression was an exploratory analysis to assess whether difference of mean age between the PCOS patients and healthy controls across different studies affects pooled effect sizes despite matching cases and controls for age within studies. We revealed that the relationship between follistatin and PCOS is modified by mean difference of age between the PCOS patients and healthy controls, suggesting that results of studies with larger difference of mean age between the PCOS patients and healthy controls should be interpreted cautiously. This was added to the manuscript (see lines 285-291).

5-Line s 189-191, “For follistatin, it was found that the relationship between follistatin and PCOS is modified by mean difference of age between the PCOS patients and healthy controls (Coefficient: - 0.06, SE: 0.01, p=0.007)”. How will you apply this result? Would you conclude that results are questionable/unreliable from those studies with larger difference of mean age between the PCOS patients and healthy controls?

Response: it shows that results of studies with larger difference of mean age between the PCOS patients and healthy controls should be interpreted cautiously. This was added to the manuscript (see lines 290-291).

6-Lines 191-193, “…by increase in the mean difference of age between the PCOS patients and healthy controls, there was a significant elevation in the mean difference of follistatin between the groups (Fig 4).” This sentence indicates a positive correlation. But from Fig. 4, we see a negative correlation,

Response: thanks for your comment. It was revised (see line 202).

Yours sincerely, 

Shahab Alizadeh, PhD.

Department of Clinical Nutrition

School of Nutritional Sciences and Dietetics

Tehran University of Medical Sciences

Fax: +98-21-889 559 79, Telephone: +98-9197604090 

Email address: sh_alizadeh@razi.tums.ac.ir

---

## [Decision Letter · Decision Letter 1]

12 Jan 2021

PONE-D-20-19835R1

Circulating resistin and follistatin levels in obese and non-obese women with polycystic ovary syndrome: a systematic review and meta-analysis

PLOS ONE

Dear Dr. alizadeh,

Thank you for submitting your manuscript to PLOS ONE. After careful consideration, we feel that it has merit but does not fully meet PLOS ONE’s publication criteria as it currently stands. Therefore, we invite you to submit a revised version of the manuscript that addresses the points raised during the review process.

please provide the funnel plots detailed in the manuscript

We look forward to receiving your revised manuscript.

Kind regards,

Stephen L Atkin, MD

Academic Editor

PLOS ONE

Reviewers' comments:

Reviewer's Responses to Questions

**Comments to the Author**

1. If the authors have adequately addressed your comments raised in a previous round of review and you feel that this manuscript is now acceptable for publication, you may indicate that here to bypass the “Comments to the Author” section, enter your conflict of interest statement in the “Confidential to Editor” section, and submit your "Accept" recommendation.

Reviewer #1: All comments have been addressed

Reviewer #2: (No Response)

2. Is the manuscript technically sound, and do the data support the conclusions?

Reviewer #1: Yes

Reviewer #2: (No Response)

3. Has the statistical analysis been performed appropriately and rigorously? 

Reviewer #1: Yes

Reviewer #2: (No Response)

4. Have the authors made all data underlying the findings in their manuscript fully available?

Reviewer #1: Yes

Reviewer #2: (No Response)

5. Is the manuscript presented in an intelligible fashion and written in standard English?

Reviewer #1: Yes

Reviewer #2: (No Response)

6. Review Comments to the Author

Reviewer #1: (No Response)

Reviewer #2: Line 137, “Publication bias was evaluated using funnel plot…”, but no funnel plots were provided. Please add funnel plots for those meta-analysis of 10 or more studies.

7. PLOS authors have the option to publish the peer review history of their article (what does this mean?). If published, this will include your full peer review and any attached files.

Reviewer #1: No

Reviewer #2: No

---

## [Author Response · Author response to Decision Letter 1]

13 Jan 2021

12 January 2021

Re:

MS title: " Circulating resistin and follistatin levels in obese and non-obese women with polycystic ovary syndrome: a systematic review and meta-analysis (Ms# 

PONE-D-20-19835)" 

Dear Editor-in-Chief of PLOS ONE

We appreciate the time and efforts by the editor and referees in reviewing our manuscript. Below are the reviewers’ verbatim comments. All changes in the manuscript have been highlighted with red font. 

We hope that with these changes and clarifications, our manuscript will be acceptable for publication. Nevertheless, we are prepared to revise our manuscript further, should it be necessary. 

COMMENTS FOR THE AUTHOR:

Reviewer #2: Line 137, “Publication bias was evaluated using funnel plot…”, but no funnel plots were provided. Please add funnel plots for those meta-analysis of 10 or more studies.

Response: funnel plots for meta-analysis of 10 or more studies were added (see Fig 8)

Yours sincerely, 

Shahab Alizadeh, PhD.

Department of Clinical Nutrition

School of Nutritional Sciences and Dietetics

Tehran University of Medical Sciences

Fax: +98-21-889 559 79, Telephone: +98-9197604090 

Email address: sh_alizadeh@razi.tums.ac.ir

---

## [Editor Report · Decision Letter 2]

15 Jan 2021

Circulating resistin and follistatin levels in obese and non-obese women with polycystic ovary syndrome: a systematic review and meta-analysis

PONE-D-20-19835R2

Dear Dr. alizadeh,

We’re pleased to inform you that your manuscript has been judged scientifically suitable for publication and will be formally accepted for publication once it meets all outstanding technical requirements.

Kind regards,

Stephen L Atkin, MD

Academic Editor

PLOS ONE
---

## [Editor Report · Acceptance letter]

11 Mar 2021

PONE-D-20-19835R2 

Circulating resistin and follistatin levels in obese and non-obese women with polycystic ovary syndrome: a systematic review and meta-analysis 

Dear Dr. Alizadeh:

I'm pleased to inform you that your manuscript has been deemed suitable for publication in PLOS ONE. Congratulations! Your manuscript is now with our production department. 

Kind regards, 

on behalf of

Dr. Stephen L Atkin 

Academic Editor

PLOS ONE